# Entirely Anatolian Hydrobiid (Caenogastropoda–Truncatelloidea–Hydrobiidae) Clade Revisited: Two More New Genera and Six New Species [note 1]

**DOI:** 10.3390/ani15172512

**Published:** 2025-08-26

**Authors:** Aleksandra Jaszczyńska, Sebastian Hofman, Deniz Anıl Odabaşı, İhsan Ekin, Ioan Sîrbu, Andrzej Falniowski

**Affiliations:** 1Department of Malacology, Institute of Zoology and Biomedical Research, Jagiellonian University, 30-387 Kraków, Poland; a.jaszczynska@uj.edu.pl; 2Department of Invertebrate Evolution, Institute of Zoology and Biomedical Research, Jagiellonian University, 30-387 Kraków, Poland; 3Department of Comparative Anatomy, Institute of Zoology and Biomedical Research, Jagiellonian University, 30-387 Kraków, Poland; s.hofman@uj.edu.pl; 4Department of Marine and Inland Water Sciences, Faculty of Marine Science and Technology, Çanakkale Onsekiz Mart University, 17100 Çanakkale, Türkiye; aodabasi@comu.edu.tr; 5Department of Energy Systems Engineering, Faculty of Engineering, Şırnak University, 73000 Şırnak, Türkiye; ekinihsan@gmail.com; 6Biology and Ecology Research Center, Faculty of Sciences, Lucian Blaga University of Sibiu, 550012 Sibiu, Romania; ioan.n.sirbu@ulbsibiu.ro

**Keywords:** integrative taxonomy, cytochrome oxidase, morphology vs. molecules, species delimitation

## Abstract

Truncatelloidea are the richest in species group of fresh- and brackish water snails, still poorly studied. The Anatolia harbours many truncatelloid species, often endangered by human activity and climate warming, especially dangerous for these snails inhabiting springs and adapted to cold water. But these snails are still hardly known. Despite the close geographic relationship between Asia Minor and Europe, some Anatolian groups have no closer relationships with the European ones; rather, relationships with the Asiatic fauna must be considered. We have studied one of such groups, finding six species new to science, and four of them representing two new genera; we describe these snails. Minute dimensions, resulting in simplified anatomy, as well as wide variation in the morphology, cause serious problems with the reconstruction of evolutionary (phylogenetic) relationships between these snails, and even with species distinction. Thus, the application of molecular data—DNA sequences—has become obligatory to solve this problem. However, sometimes the molecular data, and especially the results of their analysis, are not congruent with the morphological ones, and this is the case we are describing. This once more stresses the necessity of an integrative, holistic approach in the study of animal evolution.

## 1. Introduction

The freshwater truncatelloid fauna of Turkey (Türkiye) has been studied by some researchers [1,2,3,4,5,6,7]. Nonetheless, our current knowledge of the hydrobiid fauna inhabiting Türkiye is notably limited in comparison with what is known about it in other Mediterranean countries [8]. Türkiye encompasses three of the biogeographical regions of the Palaearctic [9]—Anatolian, Mediterranean, and Black Sea—resulting in a rich biodiversity. Considering the minute size of the hydrobiid snails, which are easily overlooked and rather hard to study, their contribution to this biodiversity should still be significantly underestimated.

To date, more than 70 hydrobiid taxa have been reported from Türkiye, the majority of which have a pattern of divergence from the west towards the southeast [5,10,11,12]. Many hydrobiid genera known to occur in Türkiye have a circum-Mediterranean distribution, such as *Pseudamnicola* Paulucci, 1878, and *Grossuana* Radoman, 1983, which probably reflects the Messinian Salinity Crisis. Others, however, are endemic to Anatolia (i.e., *Turcorientalia* Radoman, 1973; *Erosiconcha* Delicado et Gürlek, 2021; *Cariohydrobia* Delicado et Gürlek, 2021; *Anadoludamnicola* Şahin, Koca et Yıldırım, 2012) [5,8], being possibly later immigrants from Asia.

Odabaşı et al. [13] described a new monotypic genus, *Kozanium* Odabaşı et Falniowski, 2025, with its type species, *Kozanium torosum* Odabaşı et Falniowski, 2025, from Dağılcak Spring, Dağılcak Nature Park, Kozan district of Adana province. *Kozanium* molecularly clustered within the entirely Anatolian clade, with *Sheitanok* Schütt et Şeşen, 1991, type species *S. amidicus* Schütt et Şeşen, 1991 (represented by *Sheitanok* sp. in their molecular phylogeny: sequence of Delicado et al. [14]), and *Anadoludamnicola* Şahin, Koca et Yıldırım, 2012, represented by the type species *A. gloeri* Şahin, Koca et Yıldırım, 2012. Recently, we have collected truncatelloidean gastropods at five new localities in SE Anatolia. They represent six taxa new to science, all of them clustered within this entirely Anatolian clade. The study describes these new taxa.

Our study should extend the knowledge about the Turkish microsnail fauna, their diversity and habitats, stressing the urgent need for its conservation.

## 2. Materials and Methods

The snails have been collected by sieve at five sampling locations, which are springs or spring-fed streams (Table 1, Figure 1 and Figure 2) in the south and southeast of Türkiye.

Locality 1 is situated in the upper region of the Seyhan River basin. The sampling point is a wide stream called the Sarız Çayı near Yamanlı Village in the Tufanbeyli District of the Adana Province (Figure 2C). A thermal power station is located very close to the sampling point. The stream is wide but shallow and fast-flowing, and the water is rather turbid. The sampling location’s bank has sparse vegetation and has been heightened using natural riverbed soil. Locality 2 (Figure 2D) is characterized by its small stream, which is fast-flowing, clear, and cold. It is located at a high elevation in the mountainous region and flows through the village of Küçükçamurlu in the Göksun district of the Kahramanmaraş province. The riverbed is composed of stones and pebbles, with a small amount of macrophytes. The presence of small springs along the riverbed is noteworthy, and these can be seen as they line the river, creating a unique natural landscape. Locality 3 (Figure 2E,F), the Küp Şelalesi waterfall flows into the Doğan Çayı stream near Küp village in the Aladağ district of the Adana province. The material was collected from a large spring situated on the elevated western bank of the stream. This spring then discharges into the stream. Locality 4 (Figure 2A,B) and 5, Finik Castle Ruins, located in Güçlükonak town of Şırnak province, along the coast of the Dicle River. It is known that the castle was constructed in the Asur times by natural limestone (http://www.guclukonak.gov.tr/finik-surlari, accessed on 2 June 2025). The springs in the castle flow into the river, which has sparse vegetation along its banks. *Dicle bilgini* and *Sheitanok esmaae* have been sampled in the same spring.

Samples were collected by hand using tweezers from the macrolithic habitat at the sampling points (Table 1, Figure 2). For the molecular study, snails were washed twice in 80% ethanol and left to stand in this solution for ca. 12 h, after which the solution was changed twice in 24 h. Finally, after a few days, the 80% solution was exchanged for a 96% solution of ethanol, and the material was stored at −20 °C. The shells were photographed with a Canon EOS 50D digital camera (Ōita, Japan), under a Nikon SMZ18 microscope (Tokyo, Japan) with a dark field. The dissections were conducted under a Nikon SMZ18 microscope with a dark field, equipped with a Nikon DS-5 digital camera, whose captured images were used to draw anatomical structures with a graphic tablet. Morphometric parameters of the shell were measured by one person using a Nikon DS-5 digital camera (Tokyo, Japan) and ImageJ v. 1.53g image analysis software [17]. Seven biometric parameters of the shell were measured (Figure 3), following the scheme of Falniowski et al. [18]. The type materials are deposited at the Limnology Museum of Çanakkale Onsekiz Mart University (COMULM), Çanakkale, Türkiye.

Statistical multivariate analysis of biometrical variables and molecular Operational Taxonomic Units (mOTU) p-distances was conducted in Canoco 5.15 [19,20]. The p-distances between the mOTUs were subject to a Principal Coordinates Analysis (PCoA) using all axes with positive eigenvalues. The biometrical data table was analyzed using PCA (Principal Components Analysis), and a direct ordination method was performed, namely RDA (Redundancy Analysis), with the species as the explanatory variable. Response data (biometrical descriptors) were log-transformed by the relation Y′ = log(1 + Y) and centered by variables. Significance was tested with the Monte Carlo permutation test, using 999 unrestricted permutations and the level of alpha = 0.05. Polygon envelopes have illustrated the grouping of individuals, with each species being ascribed a different color. Simple and conditional term effects have tested the capacity of species to explain differences in biometrical values. To avoid errors in multiple testing, we also used adjusted *p*-values based on the false discovery rate. In the diagrams, the species are coded by the first three letters of the genus, followed by the first three letters of the species. Each code also has a mOTU ascribed, as follows: Dicbil = *Dicle bilgini* (mOTU A), Adatuf = *Adaniya tufanbeyi* (mOTU B), Adaozb = *Adaniya ozbeki* (mOTU C), Adacun = *Adaniya cuneytsolaki* (mOTU D), Sheesm = *Sheitanok esmaae* (mOTU E), Anaayh = *Anadoludamnicola ayhani* (mOTU H), Koztor = *Kozanium torosum* (mOTU I). There are some mOTU without biometry or species’ names coded as follows: *Anadoludamnicola gloeri* = mOTU G and code Anaglo (found in site F), and *Sheitanok* sp. = mOTU F and code Shesp (found in site D).

The shells and opercula were cleaned with an ultrasonic cleaner. The penes were photographed under a Motic microscope (Xiamen, China) with a dark field. The radulae were extracted with Clorox™ (Oakland, California, US), applying the techniques described by Falniowski [21], and examined and photographed using a HITACHI S-4700 scanning electron microscope (Tokyo, Japan).

DNA was extracted from whole specimens; tissues were hydrated in tris-EDTA (TE) buffer (3 × 10 min); then, total genomic DNA was extracted with the Sherlock extraction kit (A&A Biotechnology, Gdańsk, Poland), and the final product was dissolved in 20 μL of TE buffer. The extracted DNA was stored at −80 °C at the Department of Malacology, Institute of Zoology and Biomedical Research, Jagiellonian University in Kraków (Poland). Mitochondrial cytochrome oxidase subunit I (COI) was sequenced. Details of PCR conditions, primers used, and sequencing followed Szarowska et al. [22]. In the phylogenetic analysis, sequences from GenBank were used (Table 2). For analysis, we selected the sequences of snails most closely related to the newly described species, since the analysis of the biodiversity of Anatolia was not our aim. The sequences and our preliminary phylogenetic analyses were selected based on the Barcode of Life Data System [23]. All sequences were initially aligned in MUSCLE [24] implemented in MEGA 7 [25] and then checked in Bioedit 7.1.3.0 [26]. Genetic uncorrected p-distances were calculated in MEGA 7, with transitions and transversions included and uniform rates among sites. The estimation of the proportion of invariant sites and the saturation tests for entire datasets [27,28] were performed with DAMBE [29]. The data were analysed using approaches based on Bayesian inference (BI) and maximum likelihood (ML). For RAxML analysis, the jModelTest2 via the CIPRES Science Gateway [30] was used to find the best-fitting model; the model TrN+I+G was chosen. The ML analysis was conducted in RAxML-NG v. 0.8.0 [31] via the web service available at https://raxml-ng.vital-it.ch/, (accessed on 17 June 2025) with ten random and ten parsimony-started trees. In the BI analysis, the TrN+I+G model of nucleotide substitution was also applied. The analyses were run using MrBayes v.3.2.7a [32] with default priors. Two simultaneous analyses were performed, each with 10,000,000 generations, with one cold chain and three heated chains, starting from random trees and sampling the trees every 1000 generations. The first 25% of the trees were discarded as burn-in. The analyses were summarized as a 50% majority-rule consensus tree. Convergence was checked in Tracer v.1.7.1 [33]; in all cases, the Effective Sample Size exceeded 200. FigTree v.1.4.4 [34] was used to visualise the trees. Three species delimitation methods were performed: Poisson Tree Processes (PTP) [35], Automatic Barcode Gap Discovery (ABGD) [36], and Assemble Species by Automatic Partitioning (ASAP) [37]. The PTP approach was run using the web server (https://species.h-its.org/ptp/), accessed on 17 June 2025, with 100,000 MCMC generations, 100 thinning, and 0.1 burn-in. We used the RAXML output phylogenetic tree. The ABGD approach used the web server (https://bioinfo.mnhn.fr/abi/public/abgd/abgdweb.html, accessed on 17 June 2025) with the default parameters. The ASAP approach was run using the web server (https://bioinfo.mnhn.fr/abi/public/asap/, accessed on 17 June 2025), according to simple distance (p-distances). The Fastachar application [38] and DeSignate [39] were used to determine the Molecular Diagnositic Characters (MDCs) for COI. Two types of characters were accepted: at binary positions (where the character state in the query group is different from the uniform character state of the reference group) and at asymmetric positions (where the character state in the query group is not different from the uniform character state of the reference group). Molecular diagnoses for the genus were verified against all genera within a large clade (mOTU A–mOTU I). For individual species, molecular diagnoses were determined between species within a given genus. The letters used in the MDCs list denote the nucleotide position in our sequence set, and the letter in parentheses denotes the specific nucleotide characteristic for the particular group.

## 3. Results

### 3.1. Molecular Part

We obtained 12 new sequences of COI (467 bp, GenBank accession numbers PX127888-PX127899). The test for the substitution saturation analysis showed an ISS (0.75) significantly smaller than the critical ISS value (ISSC: 0.94), indicating that sequences are not saturated and thus useful in phylogenetic reconstruction. In all analyses, the topologies of the resulting phylograms were identical in both the ML and BI phylogram analyses; thus, we present the phylogram computed with RAxML.

In our molecular tree (Figure 4), all the collected gastropod taxa clustered within the well-supported (bootstrap support 90%, BP 0.94) clade, which consisted entirely of Anatolian taxa. Within this clade, there were two subclades (supports either 72 or 74%): one of them included *Sheitanok* Schütt et Şeşen, 1991, with *Sheitanok* sp. [14], *Anadoludamnicola* Şahin, Koca et Yıldırım, 2012, and *Kozanium* Odabaşı et FalniowskiOTU, 2025. One of our sequences clusters as a sister taxon with *Sheitanok* sp., another one with *Anadoludamnicola gloeri* Şahin, Koca et Yıldırım, 2012. All three techniques of species delimitation confirmed the distinctness of our two taxa. The other subclade consisted only of our new sequences, grouped in two clusters of presumably genus level, whose distinctness was confirmed by all three techniques. In the second of them, including the sequences from three localities, both ASAP and ABGD species delimitation techniques resulted in no species-level delimitation. On the other hand, bPTP delimited three species. The p-distances are given in Table 3.

### 3.2. Shell Biometry

Shell measurements are shown in Table 4, and the measurement diagram in Figure 3.

The PCoA diagram on mOTU is depicted in Figure 5. Considering proximity as an indicator of similarity, and in this case, reduced p-distance, we can distinguish broadly four groups: closely placed G and H (Anaglo and Anaayh), then in the lower left quarter, a closely related D and C (Adaozb and Adacun), followed by B (Adatuf), and further distanced from A (Dicbil). E and F share the same space in the lower right quarter (Shesp. and Sheesm), while I (Koztar) is placed isolated from all the others. The PCA of individual biometric variables classified by species is depicted in Figure 6. Most species have distinguished positions based on biometrical variables of their individuals, while some slight overlap (only at the edges of their variability) is also registered. Sheiesm, Adaozb, and Adatuf do not overlap at all, whereas the others are more or less slightly closer and partially overlap. If the species is taken as a predictor, the corresponding RDA is shown in Figure 7 (relations between biometrical variables constrained by the species in an RDA ordination diagram) and Figure 8 (classified RDA individuals diagram based on relations between biometrical descriptors predicted by the species). The RDA shows a highly significant and strong relation between the species and biometrical descriptors (test on all axes pseudo-F = 35.9, *p* = 0.001, adjusted explained variation is 80.72%). The biometrical variable alpha is maximized in Sheesm, followed by Anaayh, while the rest of the morphological descriptors are positively correlated and maximized in Anatuf. All species are well dispersed across the ordination space, hinting towards a dissimilarity and certain contrasts in terms of biometrical variables between the taxa. The relations can be traced finely in Figure 8.

We have also tested the effects of species on biometrical variability in individuals. The results are shown in Table 5.

The simple term effects (as if each species acts alone) show significant values for Adatuf, Sheesm, Adaozb, and Anaayh, while the last three species have an insignificant effect (these being also those which show a certain overlap in Figure 8). In sharp contrast, if conditional term effects are considered (the effect of each predictor after accounting for the effect of the formerly selected predictor(s)), all predictors prove to be significant. There are mostly significant differences in biometrical variability between species, but a certain overlap is also noticed at least between some taxa. The limitation is due to the reduced number of individuals analyzed in this study.

### 3.3. Systematic Part


**Truncatelloidea Gray, 1840**



**Hydrobiidae W. Stimpson, 1865**



**Horatiinae D.W. Taylor, 1966**



**Genus *Dicle***
**Falniowski and Odabaşı, n. gen.**


urn:lsid:zoobank.org:act:9CE9C090-DBBD-44CE-922E-896ABC007376

Type species ***Dicle bilgini* Falniowski and Odabaşı, n. sp.** by monophyly.


**Diagnosis**


Shell ovate-conic, broad, with spire moderately high, whorls slightly convex, body whorl massive, aperture broadly ovoid, parietal lip broad and complete, umbilicus moderately broad; female reproductive organs with one, distal receptaculum seminis and cylindrical rather bulky bursa copulatrix; penis simple, gradually tapering or broadly triangular, with a small outgrowth on its left side, its termination blunt with small papilla, vas deferens visible inside. Binary MDCs: 54(G), 234(C), 294(C), 376(C), 378(T); asymmetric MDCs: 9(A), 21(A), 195(C), 198(G), 225(C), 273(A), 429(G), 444(T), 462(A).


**Etymology**


The genus is named after the Dicle River, also known as the Tigris, which, together with the Euphrates, forms the Upper Mesopotamia region in southeastern Türkiye.


***Dicle bilgini* Falniowski and Odabaşı, n. sp.**


urn:lsid:zoobank.org:act:86D8F90A-2B55-40CA-B556-68041CA04ACC

GenBank no. for COI: PX127896-PX127899

Type locality: Spring in Finik Harabeleri (Finik Ruins), Eskiyapı, Güçlükonak, Şırnak, Türkiye (locality 4, Figure 2A,B).

Holotype: (Figure 9A) Ethanol-fixed specimen, Spring in Finik Harabeleri (Finik Ruins), Eskiyapı, Güçlükonak, Şırnak, Türkiye (locality 4), leg. İhsan Ekin, 06.04.2025. Voucher number COMULM-G 302.

Paratypes: 73 ethanol-fixed specimens, Spring in Finik Harabeleri (Finik Ruins), Eskiyapı, Güçlükonak, Şırnak-Türkiye, Voucher number COMULM-G 303.

**Diagnosis** As for the genus


**Description**


*Shell* (Figure 9A,E)

Up to 1.87 mm high, ovate-conic, broad, with a moderately high spire, about four whorls, and a spire height of 22–24% of the shell height. Teleoconch whorls slightly convex, growing regularly in diameter, suture shallow but well-marked, body whorl massive. Aperture broadly ovoid, outer lip simple, parietal lip broad and complete, umbilicus moderately broad. Teleoconch moderately thick-walled, its wall translucent, with delicate growth lines, periostracum whitish or yellowish. Shell measurements: Table 4. Operculum paucispiral, oval, smooth on both surfaces.

*Radula* (Figure 10H). The central tooth has short and blunt cusps according to the following formula:(5)4–1–4(5)1–1

The median cusp about three times longer than the adjacent ones, a deep sinus at the frontal side of the cutting edge, and the basal tongue is narrowly V-shaped. The lateral tooth with 2–1–2(3) blunt cusps; the biggest one is slightly longer than the adjacent ones, massive, and with a rounded tip. The inner marginal tooth with 22–24 long and sharp cusps, and the outer marginal one with about 18 sharp cusps.


*Morphology and anatomy of soft parts*


The head, tentacles, and mantle are intensely pigmented. There is no ctenidium. The female reproductive organs with one distal receptaculum seminis and a cylindrical, bulky bursa copulatrix. The penis (Figure 11A–F) simple, gradually tapering or broadly triangular, with a small outgrowth on its left side; its termination blunt with a small papilla, and the vas deferens visible inside.


**Etymology**


The specific epithet *bilgini* refers to Fikret Bílgín, estimable malacologist deeply devoted to the study of the Turkish molluscs.


**Known distribution**


Also in the Spring in Finik Harabeleri (Finik Ruins), Eskiyapı, Güçlükonak, Şırnak-Türkiye (locality 5), about 500 m from the type locality.


**Differential diagnosis**


The radula in *Dicle bilgini* with a central tooth with fewer cusps on the cutting edge and less prominent basal cusps than in *Kozanium*, as well as in *Adaniya* n. gen. Unlike in *Sheitanok*, the shell is not valvatoid. *Anadaluamnicola* differs in the lack of basal cusps. In the female reproductive organs of *Dicle bilgini,* the bursa copulatrix is cylindrical, unlike in the other genera within this clade. The penis, unlike *Anudaluamnicola,* is never strap-like, and unlike *Sheitanok,* without a big terminal papilla, is covered with a thin layer of chitin. In *Adaniya,* there is either no/vestigial or big and broad outgrowth on its left side.


**Genus *Adaniya* Odabaşı**
** and Falniowski gen. nov.**


urn:lsid:zoobank.org:act:0C9EFD28-94E2-4BD2-A689-1D726C953567

Type species ***Adaniya tufanbeyi* Jaszczyńska and Odabaşı n. sp.**


**Diagnosis**


Shell ovate-conic or trochiform, with spire low or moderately low, body whorl massive, aperture broadly ovoid, parietal lip broad and complete, umbilicus moderately broad; or broad; female reproductive organs with one, distal receptaculum seminis and moderately big or small bursa copulatrix; penis broadly triangular, simple or with a lobe on its left side. Binary MDCs: 288(C), 345(C), 351(T), 363(G), 423(G); asymmetric MDCs: 90(A).


**Etymology**


This genus is named after Adaniya, the name by which the region was known during the time of the Hittite civilization in Anatolia.


**Differential diagnosis**


Unlike in *Sheitanok*, the shell is not valvatoid. Unlike *Anudaloamnicola*, there are basal cusps on the central tooth in the radula. The penis, unlike *Anudaluamnicola,* is never strap-like, and unlike *Sheitanok,* it lacks a big terminal papilla covered with a thin layer of chitin. Bursa copulatrix and especially receptaculum seminis are smaller than those in *Kozanium.*


**
*Adaniya tufanbeyi*
**
** Jaszczyńska and Odabaşı n. sp.**


urn:lsid:zoobank.org:act:0774F308-2E86-4539-B719-8A91E51AC1A0

GenBank no. for COI: PX127892

Type locality: Sarız Çayı, Near Yamanlı Village, Tufanbeyli District, Adana Province, Türkiye (locality 1, Figure 2C)

Holotype: (Figure 9F) Ethanol-fixed specimen, Sarız Çayı, Near Yamanlı Village, Tufanbeyli District, Adana Province, Türkiye, leg. Deniz Anıl Odabaşı, 19.10.2024. Voucher number COMULM-G 304.

Paratypes: 29 ethanol-fixed specimens, Sarız Çayı, Near Yamanlı Village, Tufanbeyli District, Adana Province, collection of the COMULM (Voucher number COMULM-G 305), Türkiye.


**Diagnosis**


A representative of Adaniya with an ovate-conical, broad shell; a central tooth with a massive cutting edge and about ten minute cusps; two pairs of basal cusps; a bursa copulatoria whose duct is not sharply demarcated from the bursa; and a penis broadly triangular, with or without a very small outgrowth on its left side. Binary MDCs: 21(T), 84(T), 88(T), 99(G), 117(T), 177(T), 255(G), 258(A), 300(A), 402(A).


**Description**


*Shell* (Figure 9F,G)

Up to 3.12 mm high, ovate-conic, broad, with a moderately high spire, about four whorls, and a spire height of 26–28% of the shell height. Teleoconch whorls slightly convex, growing regularly in diameter, suture shallow but well-marked, body whorl massive. Aperture broadly ovoid, near circular in shape, outer lip simple, parietal lip narrow and complete, umbilicus slit-like. Teleoconch rather thick-walled, growth lines nearly invisible, periostracum yellowish. Shell measurements: Table 4. Operculum paucispiral, oval, smooth on both surfaces.

*Radula* (Figure 10F) The central tooth has short and blunt cusps according to the following formula:(10)9–1–9(10)2–2

The median cusp broad and massive, about three times longer than the small, sharp adjacent cusps. A deep sinus present on the front side of the cutting edge, and the basal tongue narrowly V-shaped. The lateral tooth with 6–1–5 cusps; the largest cusp twice as long as the adjacent cusps and massive with a rounded tip. The inner marginal tooth with 34–36 long, sharp cusps; the outer marginal tooth with about 22 sharp cusps.


*Morphology and anatomy of soft parts*


The head and mantle intensely pigmented black. The female reproductive organs (Figure 12A) with short, broad pallial glands; a narrow loop of the oviduct; a large distal receptaculum seminis with a short duct; and a small bursa copulatrix gradually narrowing into a long duct. The penis (Figure 11G–J) broadly triangular and simple. sometimes with a very small outgrowth on its left side. Its distal end blunt with a papilla and a visible vas deferens.


**Etymology**


The specific epithet *tufanbeyi* refers to Osman Tufan Bey, who is renowned for his invaluable service to the Cilician region (today the Adana and Osmaniye provinces) during the Turkish struggle for independence. He displayed great courage and bravery in the face of adversity.

Known distribution: Type locality only.


**Differential diagnosis**


Shell much bigger than in *A. cuneytsolaki* and with a higher spire than in *A. ozbeki*. The dens centralis in the radula, unlike *A. cuneytsolaki*, with a massive cutting edge similar to that in *A. ozbeki*, but with fewer minute cusps and one, not two pairs of basal cusps. The bursa copulatrix sac-shaped without a sharply demarcated duct, unlike the other two species. The penis similar to that in *A. cuneytsolaki* but different from that in *A. ozbeki* in lacking a big triangular nonglandular lobe on its left side.


**
*Adaniya ozbeki*
**
**Hofman**
**and**
**Odabaşı n. sp.**


urn:lsid:zoobank.org:act:91BCA77D-0B27-40E6-AD60-7945B26CF166

GenBank no. for COI: PX127891

Type locality: Küçükçamurlu Spring, Göksun District, Kahramanmaraş Province, Türkiye (locality 2, Figure 2D).

Holotype: (Figure 9H) Ethanol-fixed specimen, Küçükçamurlu Spring, Göksun District, Kahramanmaraş Province, Türkiye (locality 2). leg. Deniz Anıl Odabaşı, 19.10.2024. Voucher number COMULM-G 306

Paratypes: 19 ethanol-fixed specimens, Küçükçamurlu Spring, Göksun District, Kahramanmaraş Province, collection of the COMULM (Voucher number COMULM-G 307), Türkiye.


**Diagnosis**


A representative of *Adaniya* with a trochiform and broad shell, a central tooth with a massive cutting edge, and about six minute cusps, and one pair of basal cusps. The bursa copulatrix big and oval with a short, broad duct. The penis with a wide, triangular, nonglandular outgrowth on the medial section of the left. Binary MDCs: 144(A), 147(G), 150(G).


**Description**


*Shell* (Figure 9H,I)

Up to 2.44 mm high, trochiform, and very broad. It has a low spire with about four whorls, and the spire is 12–17% of the shell’s height. The teleoconch whorls slightly convex and growing abruptly in diameter. The suture shallow and slightly marked. The body whorl very massive. Aperture broadly ovoid, near circular in shape, outer lip simple, parietal lip broad and complete, umbilicus rather broad. Teleoconch rather thick-walled, growth lines nearly invisible, periostracum yellowish. Shell measurements: Table 4. Operculum paucispiral, oval, smooth on both surfaces.

*Radula* (Figure 10C,D) The central tooth has short and blunt cusps according to the following formula:(6)5–1–5(6)1–1

The median cusp broad and massive, about four times longer than the small, blunt adjacent ones. A deep sinus present on the front side of the cutting edge, and the basal tongue narrowly V-shaped. The lateral tooth with 6–1–5 cusps; the largest cusp twice as long as the adjacent cusps and is massive with a rounded tip. The inner marginal tooth with 32–34 long, sharp cusps; the outer marginal tooth has about 23 sharp cusps.


*Morphology and anatomy of soft parts*


The head and mantle intensely pigmented black. The female reproductive organs (Figure 12B) short pallial glands, a large oval bursa copulatrix with a short, broad duct, a large, spherical distal receptaculum seminis, and a long, narrow loop of the renal oviduct. The penis (Figure 11K–N) broadly triangular with a wide, nonglandular outgrowth on the left side of the medial section. The distal end of the penis blunt with a papilla and a visible vas deferens, and some brown pigment spots on its dorsal side.


**Etymology**


The specific epithet *ozbeki* refers to Prof. dr Murat Özbek, who is one of the world’s leading experts on freshwater amphipods in Türkiye.

Known distribution: Type locality only.


**Differential diagnosis**


Shell much bigger than in *A. cuneytsolaki* and with a lower spire than in *A. tufanbeyi*. Radula differs from *A. tufanbeyi* in the central tooth with six (instead of ten) and smaller cusps along a massive cutting edge, and one, not two pairs of basal cusps; in *A. cuneytsolaki,* the cutting edge consists of fewer but much bigger cusps. Bursa copulatrix with a well-demarcated duct, unlike in *A. tufanbeyi*, and shorter and bulkier than in *A. cuneytsolaki*. Penis with a wide triangular, nonglandular outgrowth on the left part in its medial section, unlike *A. tufanbeyi* and *A. cuneytsolaki*.


**
*Adaniya cuneytsolaki*
**
**Odabaşı and Jaszczyńska n. sp.**


urn:lsid:zoobank.org:act:1833F87E-957B-45B3-9B72-8AD8BEC90B8C

GenBank no. for COI: PX127888-PX127889

Type locality: Küp Şelalesi, Doğan Çayı, Aladağ District, Adana Province, Türkiye (locality 3, Figure 2E,F).

Holotype: (Figure 9J) Ethanol-fixed specimen, Küp Şelalesi, Doğan Çayı, Aladağ District, Adana Province, Türkiye (locality 3), leg. Deniz Anıl Odabaşı, 14.10.2024. Voucher number COMULM-G 308.

Paratypes: Five ethanol-fixed specimens, Küp Şelalesi, Doğan Çayı, Aladağ District, Adana Province, collection of the COMULM (Voucher number COMULM-G 309), Türkiye.


**Diagnosis**


A representative of *Adaniya* that has a small, ovate-conical, and narrow shell; a central tooth with a non-massive cutting edge and long, sharp cusps; a sac-shaped bursa copulatrix with a short, broad duct; and a penis without any outgrowths. Binary MDCs: 66(C), 135(G), 148 (T), 198(C), 225(T), 232(T), 252(A), 414(C).


**Description**


*Shell* (Figure 9J–L)

Up to 1.83 mm high, ovate-conic, narrow, with a low or moderately high spire, with about 3½ whorls, spire height 18–26% of the shell height. Teleoconch whorls slightly convex, growing regularly in diameter, sometimes abruptly (Figure 9K), resulting in a massive body whorl, suture shallow and not well-marked. Aperture broadly ovoid, or near circular in shape, more or less dragged away from body whorl, resulting in umbilicus in a form of broad trough, outer lip simple and thick, parietal lip broad and complete. Teleoconch moderately thick-walled, growth lines slightly visible, periostracum whitish or yellowish. Shell measurements: Table 4. Operculum paucispiral, elongate-ellipsoidal, smooth on both surfaces.

*Radula* (Figure 10G) The central tooth has long and sharp cusps according to the following formula:(5)4–1–4(5)      or      4–1–41–1                         1–1

The median cusp less than twice the length of the adjacent cusps, a deep sinus on the front side of the cutting edge, and a narrow, V-shaped basal tongue. The lateral tooth with four to five sharp cusps. The largest cusp twice as long as the adjacent ones and broad with a sharp tip. The inner marginal tooth with about 26 long, sharp cusps; the outer marginal tooth with about 20 sharp cusps.


*Morphology and anatomy of soft parts*


The head and mantle intensely pigmented black. The female reproductive organs (Figure 12C) with short pallial glands, a big, elongated, sac-shaped bursa copulatrix with a short, broad duct, a moderately big, sac-shaped distal receptaculum seminis, and a short loop of the renal oviduct. The penis (Figure 13A,B) simple and broadly triangular without any outgrowths, terminated with a papilla, and the vas deferens visible inside.


**Etymology**


The specific epithet *cuneytsolaki* refers to Prof. and Dr. Cüneyt Nadir Solak, a friendly person and one of Türkiye’s leading experts on diatoms.

Known distribution: Type locality only.


**Differential diagnosis**


The shell much smaller and slimmer than in *A. tufanbeyi* and *A. ozbeki*. Radula differs from those of these two species in that the cutting edge of the central tooth has a few big, sharp cusps. Bursa copulatrix differs from *A. tufanbeyi* in having a well-discernible duct and from *A. ozbeki* in being longer and bulkier. Penis without any outgrowth, similar to that in *A. tufanbeyi*, but unlike *A. ozbeki*.


**Genus *Sheitanok***
**Schütt & Şeşen, 1991**


Type species *Sheitanok amidicus* Schütt & Şeşen, 1991


**
*Sheitanok esmaae*
**
**Ekin & Odabaşı, n. sp, 2025**


urn:lsid:zoobank.org:act:5F56C1BC-5A56-4F41-A438-55A2EEF94FCF

GenBank no. for COI: PX127893-PX127895

Type locality: Spring in Finik Harabeleri (Finik Ruins), Eskiyapı, Güçlükonak, Şırnak, Türkiye (locality 4, Figure 2A,B)

Holotype: (Figure 9M–O) Ethanol-fixed specimen, Spring in Finik Harabeleri (Finik Ruins), Eskiyapı, Güçlükonak, Şırnak, Türkiye (locality 5), leg. Ihsan Ekin, 06.04.2025. Voucher number COMULM-G 310.

Paratypes: 33 ethanol-fixed specimens, Spring in Finik Harabeleri (Finik Ruins), Eskiyapı, Güçlükonak, Şırnak, collection of the COMULM (Voucher number COMULM-G 311), Türkiye.


**Diagnosis**


A representative of the genus *Sheitanok* with a less wide umbilicus and aperture of the shell than in the type species. Binary MDCs: 12(C), 15(T), 36 (C), 40(C), 51(A), 75(C), 78(A), 96(A), 153(T), 168(T), 169(T), 171(G), 172(C), 177(T), 198(C), 201(A), 219(A), 225(G), 241(T), 270(C), 282(C), 285(A), 339(T), 396(A), 408(T), 411(T), 420(T), 429(C).


**Description**


*Shell* (Figure 9M–R)

Up to 1.27 mm high, valvatiform (depressed trochiform), broad, with spire extremely low, with about 3½ whorls, spire height 9–10% of the shell height. Teleoconch whorls flat and grow regularly in diameter. Suture shallow and not well-marked. The umbilicus moderately broad, completely open, and circular in shape. The body whorl broad and massive. The aperture narrowly ovoid with an angle at its upper side. The outer lip sinuate, and the parietal lip broad and complete. The teleoconch moderately thick-walled and translucent with slightly visible growth lines. The periostracum whitish or yellowish. Shell measurements: Table 4. Operculum paucispiral, oval with an angle, orange, smooth on both surfaces.

*Radula* (Figure 10E) The central tooth with long and blunt cusps according to the following formula:


5–1–5



(2)1–1(2)

The median cusp narrow and less than twice the length of the adjacent cusps. A deep sinus present on the front side of the cutting edge, and the basal tongue narrowly V-shaped. The lateral tooth with five blunt cusps, the largest of which slightly longer than the others and with a blunt tip. The inner marginal tooth with 24–26 long cusps, and the outer marginal tooth with about 21 cusps.


*Morphology and anatomy of soft parts*


Head and tentacles delicately pigmented, big eyespots, mantle intensely pigmented, ctenidium small with about nine filaments. Bursa copulatrix narrow and tubular in shape, with a relatively short duct and one distal seminal receptacle. Penis (Figure 13C–I) gradually tapering, simple, without outgrowths, with a big terminal papilla covered with a thin layer of chitin.


**Etymology**


The specific epithet *esmaae* refers to the late mother of Prof. and Dr. İhsan Ekin, Esma Ekin.

Known distribution: Also in the Spring in Finik Harabeleri (Finik Ruins), Eskiyapı, Güçlükonak, Şırnak, Türkiye (locality 4).


**Differential diagnosis**


Shell with a less wide umbilicus and a less wide aperture than in *S. amidicus*. Female reproductive organs similar to those described and illustrated by Schütt and Şeşen [15] for that species. Penis is also similar, as long as a schematic drawing of Schütt and Şeşen [15] is considered.


**Remarks**


*Sheitanok amidicus* was described from “Quelle im Ort Tüllük, 30 km N Diyarbakir an der Straße nach Elazlg” [15], at a straight-line distance of 190 km from our locality. The locality of Delicado et al. [14]’s *Sheitanok* sp.—“Buhur in Mardin province” does not exist; it is most probably Batur, a village in Dageçit Town in Mardin province—is 70 km from the type locality of *Sheitanok amidicus* and 180 km from our locality. Our sequences are evidently distinct from the ones of Delicado et al. [14], representing two distinct species. However, the species assignment of *Sheitanok* at both localities remains doubtful, as long as the specimens from the type locality remain unsequenced.


**Genus *Anadoludamnicola* Şahin, Koca & Yıldırım, 2012**


The type species *A. gloeri* Şahin, Koca & Yıldırım, 2012


**
*Anadoludamnicola ayhani*
**
**Odabaşı n. sp.**


urn:lsid:zoobank.org:act:4F4ABD8C-5A2F-4BAF-9F72-E7B76CCCE334

GenBank no. for COI: PX127890

Type locality: Küp Şelalesi, Doğan Çayı, Aladağ District, Adana Province, Türkiye (locality 3, Figure 2E,F).

Holotype: (Figure 9S) Ethanol-fixed specimen, Küp Şelalesi, Doğan Çayı, Aladağ District, Adana Province, Türkiye (locality 3), leg. Deniz Anıl Odabaşı, 14.10.2024. Voucher number COMULM-G 312.

Paratypes: Five ethanol-fixed specimens, Küp Şelalesi, Doğan Çayı, Aladağ District, Adana Province, collection of the COMULM (Voucher number COMULM-G 313), Türkiye.


**Diagnosis**


Representative of the genus *Anudaluamnicola* with no basal cusps on the central tooth. Binary MDCs: 9(C), 15(A), 18 (T), 69(C), 159(T), 163(T), 165(G), 169(C), 171(T), 172(T), 186(T), 195(G), 207(T), 222(C), 249(T), 265(G), 267(T), 268(C), 273(C), 289(C), 300(C), 330(C), 393(G), 405(T), 414(C), 432(T), 439(C), 462(T).


**Description**


*Shell* (Figure 9S,T)

Up to 1.66 mm high, ovate-conic, broad, with a rather low spire, about four whorls, and a spire height of 18–19% of the shell height. Teleoconch whorls nearly flat, growing regularly in diameter, with a shallow, not well-marked suture, and the body whorl massive. Aperture broadly ovoid, outer lip simple, parietal lip complete, narrow or broad, umbilicus broad. Teleoconch rather thick-walled, growth lines delicate, periostracum whitish or yellowish. Shell measurements: Table 4. Operculum paucispiral, oval, smooth on both surfaces.

*Radula* (Figure 10A,B)

The central tooth typically taenioglossate, with long and blunt cusps on the cutting edge, following the formula (6)5–1–5(6), but without the basal cusps. The median cusp narrow and less than twice the length of the adjacent cusps. A deep sinus present on the front side of the cutting edge, and the basal tongue narrowly V-shaped. The lateral tooth with (5)4–1–5 cusps; the largest cusp nearly twice as long as the adjacent cusps and narrow. The inner marginal tooth with 30–32 long cusps, and the outer marginal tooth about 16 cusps.

Morphology and anatomy of soft parts

The female reproductive organs (Figure 12D) with short pallial gland complexes, a large, spherical bursa copulatrix with a distinct, short duct on a medium-sized, sac-shaped distal receptaculum seminis, and a wide loop of the renal oviduct. The penis (Figure 13J, K) strap-like, narrow, and simple with a pointed tip and a spot of brown pigment on its dorsal side, the vas deferens visible inside.


**Etymology**


The specific epithet *ayhani* refers to the father of one of the authors, Ayhan Odabaşı.


**Known distribution**


Known from the type locality only.


**Differential diagnosis**


The radula of *Anadoludamnicola ayhani* has no basal cusps reported for *A. gloeri* by Şahin et al. [16]. The bursa copulatrix spherical, not flattened, and the receptaculum seminis shorter than in *A. gloeri*. The penis of *A. ayhani* in general resembles the one figured and described by Şahin et al. [16] for *A. gloeri*, but their description and schematic drawing make an impossibly more detailed comparison.


**Remarks**


*Anadoludamnicola gloeri* was described from İnek Pınarı just above the Tohma Çayı (Akçadağ, Malatya), at a straight-line distance of 240 km from our locality. The locality of Delicado et al. [14], Spring in Yarıkkaya, Tunceli, was 160 km from the type locality, and 390 km from our locality with *Anadoludamnicola*. Our sequences are evidently distinct from the ones of Delicado et al. [14], representing two distinct species. However, the species assignment of *Anadoludamnicola* at both localities remains doubtful, as long as the specimens from the type locality remain unsequenced.

Table 6 summarizes the diagnostic features of the newly described species.

## 4. Discussion

There is a rich literature demonstrating the need, or even necessity, of molecular data for species distinction in the Truncatelloidea (see Falniowski [54] for review), despite the fact that the molecular- and morphology-based phylogeny may differ among all the organisms [55]. In our case, there is no conflict in the resulting phylogeny, but in the species delimitation. Two of the three species delimitation techniques suggested the *Adaniya* genus to be a monotypic one, but the morphology of the shell (also biometry), radulae, female reproductive organs, and penes clearly demonstrated the distinctness of these three species. It is noteworthy that such morphological distinctness between congeners is rather rare in the Truncatelloidea [54]. The algorithmic techniques of species delimitation based on sequence data are very helpful but cannot be uncritically used—their data interpretation should be careful, and a holistic, integrative approach should be strongly recommended [14]. Similar conclusions were presented by Duminil and Di Michele for plants [56], Tödter et al. for Hydrozoa [57], and Lefébure et al. for Crustacea [58].

The basal cusps, typical of most of the Truncatelloid taxa, we have not found in *Anadoludamnicola.* Prominent basal cusps (one pair) were figured by Şahin et al. [16]. Basal cusps are absent in some groups of the Truncatelloidea, like the Emmericiidae [59].

All the species described above belong to a hydrobiid clade entirely restricted to Anatolia (or, perhaps, West Asia), having no close relatives in Europe, as already noted by Odabaşı et al. [13]. As stated, the case already seems interesting, since the isolation of Anatolia from the Balkans was neither continuous over time, nor efficient, and most of the hydrobiid genera are represented in both areas [1,22,60,61,62,63]. On the other hand, according to Gürlek et al. [5], as many as 60% of the Anatolian prosobranch species are endemic. However, the distinctiveness of many hydrobioid species remains doubtful [54].

Neubauer and Wesselingh [63] reported a high percentage of the endemic gastropods in the early Pleistocene, but of these, about half of the species were endemic to the Aegean–Anatolian region, including also mainland Greece. In their study on freshwater crabs, Ghanavi et al. [64] reported that their genetic diversity was represented by East Mediterranean and Levant groups, the latter in East Anatolia, where the type locality of our new species is located. They inferred the centre of distribution for the *Potamon* Savigny, 1816 crabs to lie in their Levant region, and this also may be the case for our gastropod clade. According to Kosswig [65], there was a migration of Asiatic as well as Indo-African (from India) freshwater fauna to Anatolia during the Pliocene. In the Pleistocene, Anatolia served as a refugium during glaciations.

## 5. Conclusions

The discovery of six new species to science, part of which are classified to two new genera, following the collection of snails in five springs, clearly demonstrates the poor state of knowledge about the fauna of minute gastropods in Turkey. Our study also confirms, once more, the necessity of applying molecular data—DNA sequences—not only to infer phylogenetic relationships, but also to discriminate species. However, as we have demonstrated, the molecular discrimination alone may be misleading, and the morphological characters should always be considered in taxonomical decisions on the species level.

## Figures and Tables

**Figure 1 animals-15-02512-f001:**
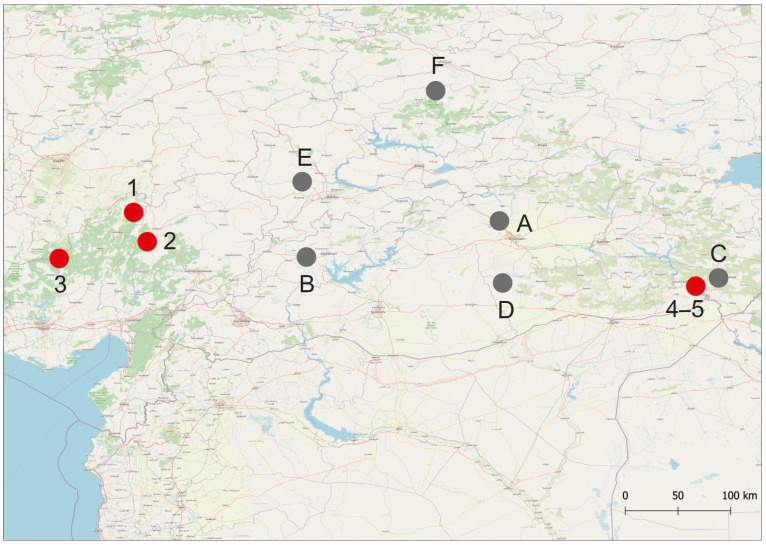
Sampling locations used in this study (red dots: 1–5, see Table 1). Some reference locations (black dots) are also shown: A–type locality of *Sheitanok amidicus* [15]; B,C–other localities of *Sheitanok amidicus* [15]; D–*Sheitanok* sp. [14]; E–type locality of *Anadoludamnicola gloeri* [16]; F–*Anadoludamnicola gloeri* [14].

**Figure 2 animals-15-02512-f002:**
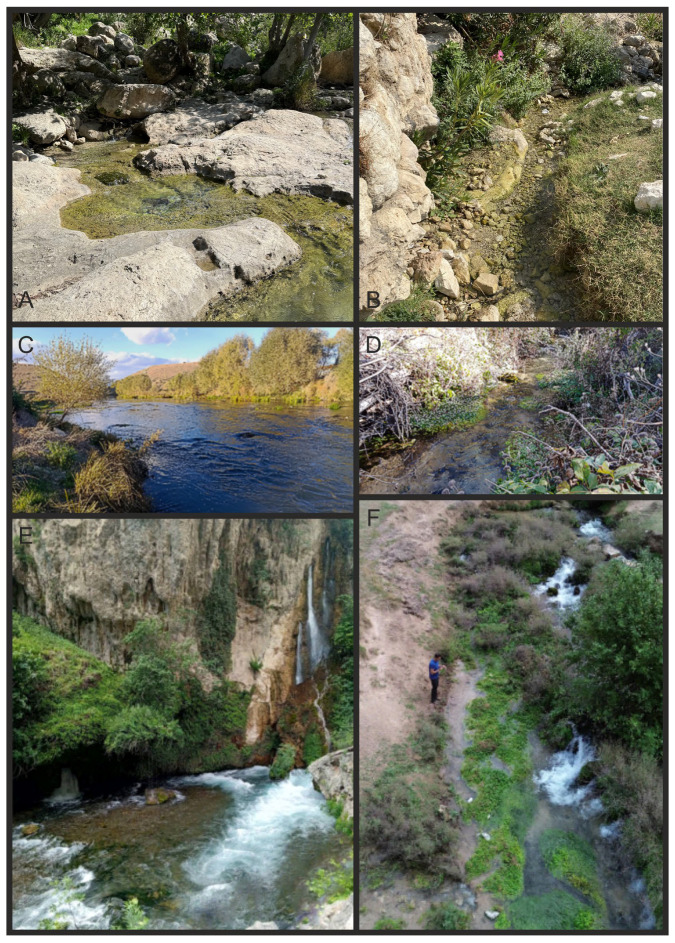
Sampled localities: (**A**,**B**)—Spring in the Finik Castle Ruins (locality 4 and 5, respectively); (**C**)—Sarız Çayı (locality 1); (**D**)—Küçükçamurlu Spring (locality 2); (**E**)—Küp Şelalesi waterfall, and the stream (**F**)—Spring flowing into the Doğan Çayı near the Küp Şelalesi waterfall (locality 3).

**Figure 3 animals-15-02512-f003:**
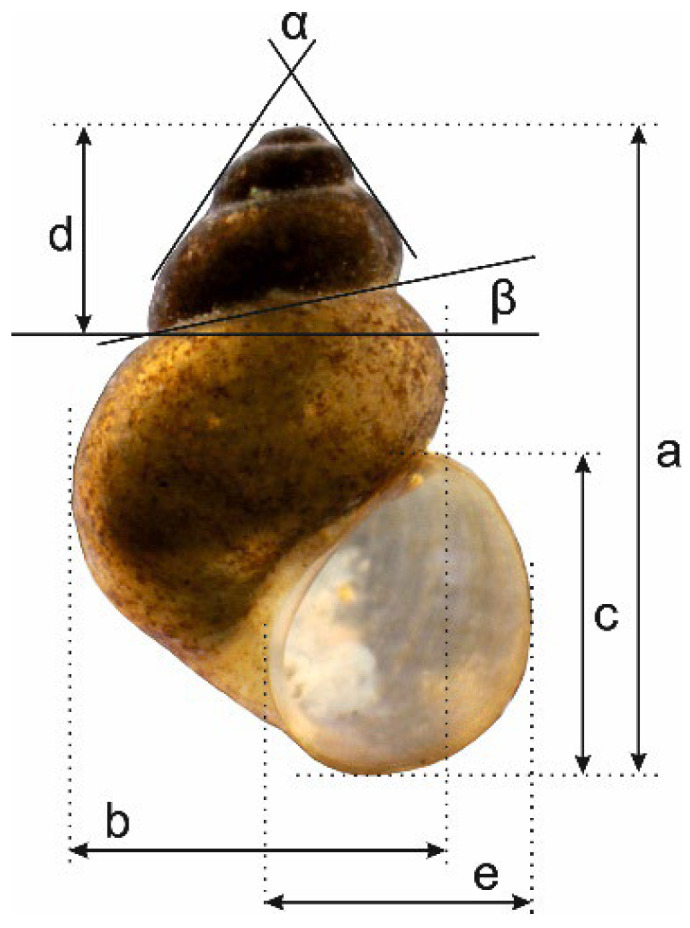
Diagram showing the measurement of the shell. Measured variables: a—shell height; b—body whorl width; c—aperture height; d—spire height; e—aperture width; α—apex angle measured between the lines tangential to the spire; β—angle between the body whorl suture and the line perpendicular to the columella.

**Figure 4 animals-15-02512-f004:**
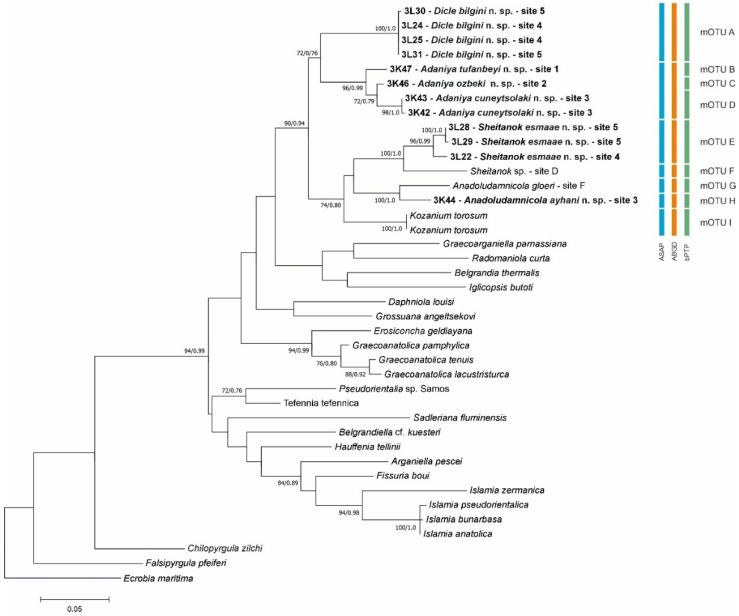
Maximum likelihood tree based on COI sequences. Bootstrap and BP for the nodes are given. Results of species delimitation are also shown. Bold indicates new sequences.

**Figure 5 animals-15-02512-f005:**
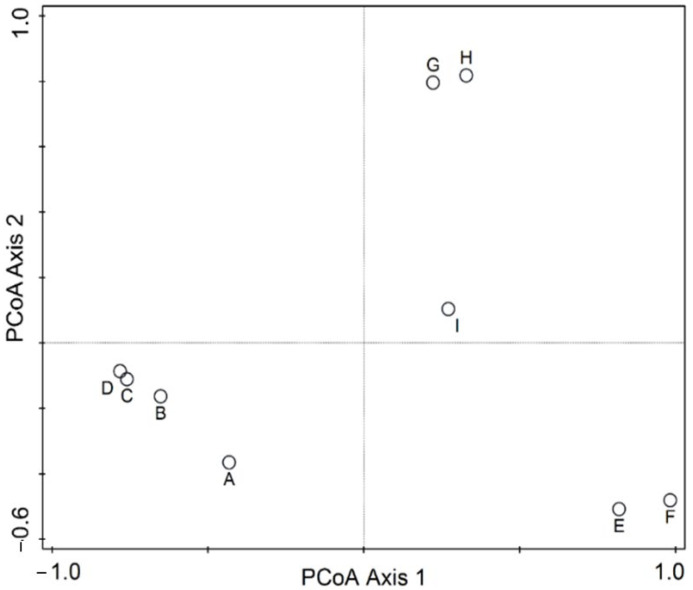
Principal Coordinate Analysis (PCoA) on the genetic p-distances between the mOTUs.

**Figure 6 animals-15-02512-f006:**
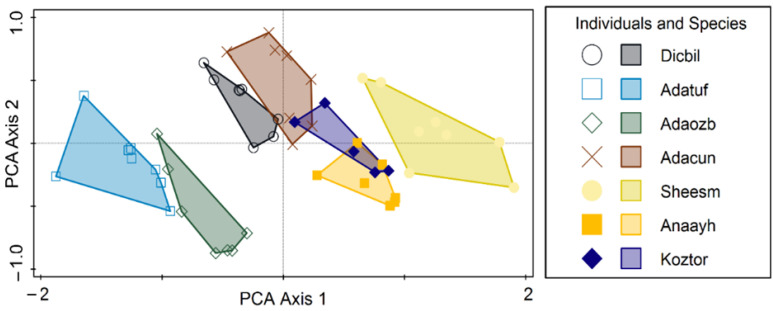
Principal Component Analysis (PCA) of individuals’ biometrical variables classified by species.

**Figure 7 animals-15-02512-f007:**
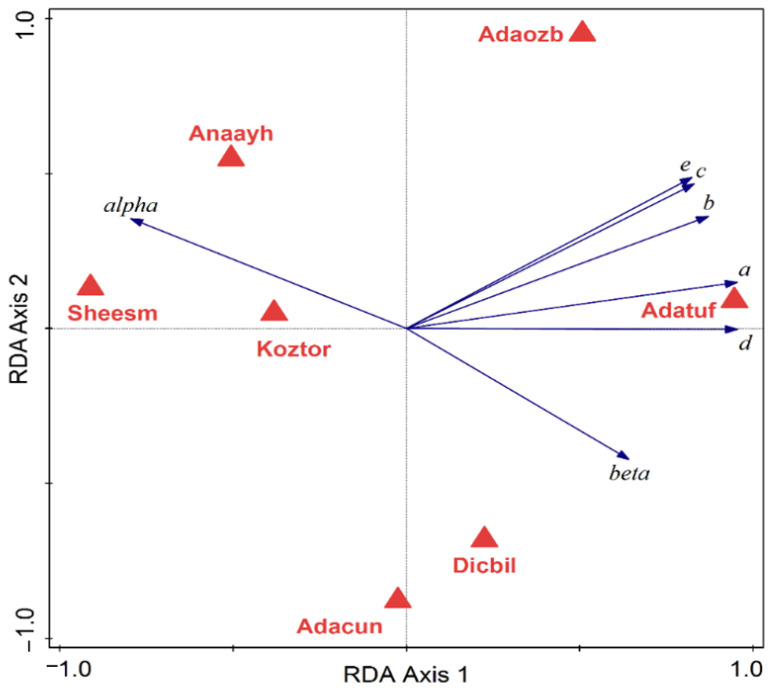
The relation between biometrical variables constrained by the species in a Redundancy Analysis (RDA) ordination diagram.

**Figure 8 animals-15-02512-f008:**
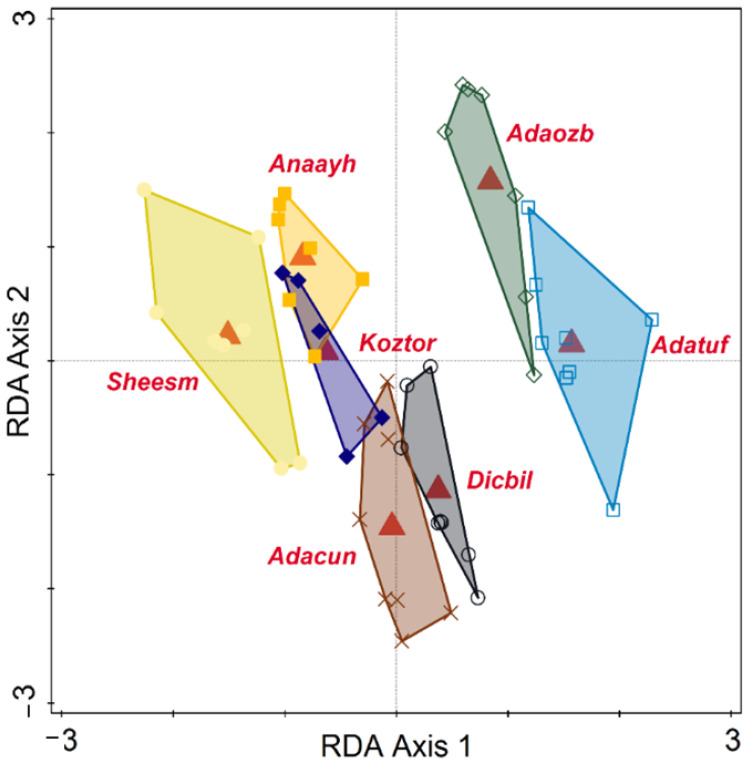
Classified RDA diagram of individuals based on relations between biometrical descriptors predicted by the species.

**Figure 9 animals-15-02512-f009:**
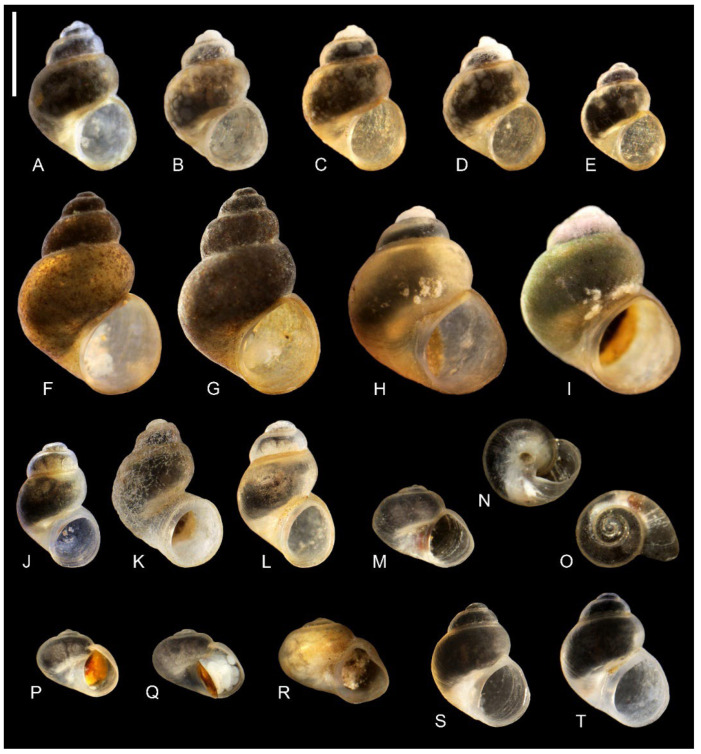
Shells of the studied taxa. (**A**–**E**)—*Dicle bilgini* n. sp.: (**A**)—holotype, (**B**)—3L30, (**C**)—3L24, (**D**)—3L25, (**E**)—3L31; (**F**,**G**)—*Adaniya tufanbeyi* n. sp.: (**F**)–holotype, (**G**)—3K47; (**H**,**I**)—*Adaniya ozbeki* n. sp.: (**H**)—holotype, (**I**)—3K46; (**J**–**L**)—*Adaniya cuneytsolaki* n. sp.: (**J**)—holotype, (**K**)—3K42, (**L**)—3K43; (**M**–**R**)—*Sheitanok esmaae* n. sp.: (**M**–**O**)—holotype, (**P**)—3L28, (**Q**)—3L29, (**R**)—3L22; (**S**,**T**)—*Anadoludamnicola ayhani* n. sp.: (**S**)—holotype, (**T**)—3K44. Scale bar: 1 mm.

**Figure 10 animals-15-02512-f010:**
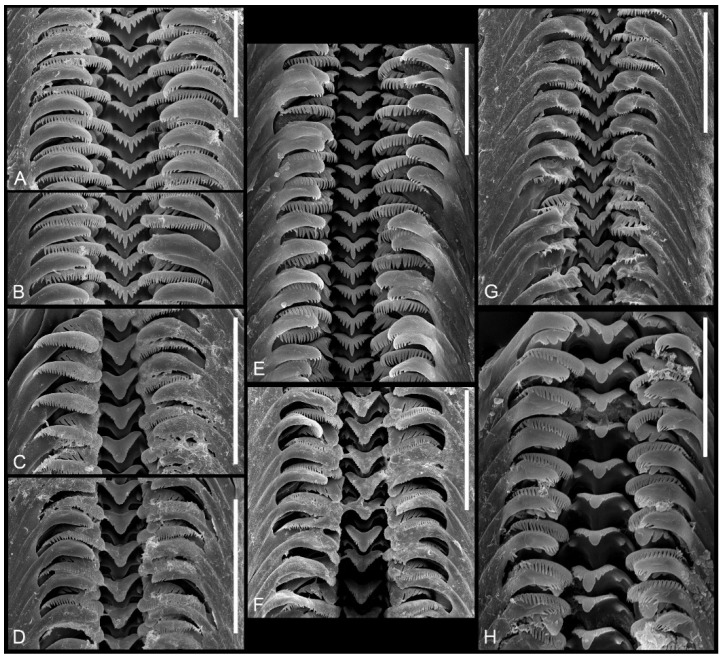
Radulae of the studied taxa. (**A**,**B**)—*Anadoludamnicola ayhani* n. sp. (locality 3); (**C**,**D**)—*Adaniya ozbeki* n. sp (locality 2); (**E**)—*Sheitanok esmaae* n. sp. (locality 4); (**F**)—*Adaniya tufanbeyi* n. sp. (locality 1); (**G**)—*Adaniya cuneytsolaki* n. sp. (locality 3); (**H**)—*Dicle bilgini* n. sp. (locality 4). Scale bar: (**A**,**B**,**E**)—20 μm; (**G**)—30 μm; (**H**)—40 μm, (**C**,**D**,**F**)—50 μm.

**Figure 11 animals-15-02512-f011:**
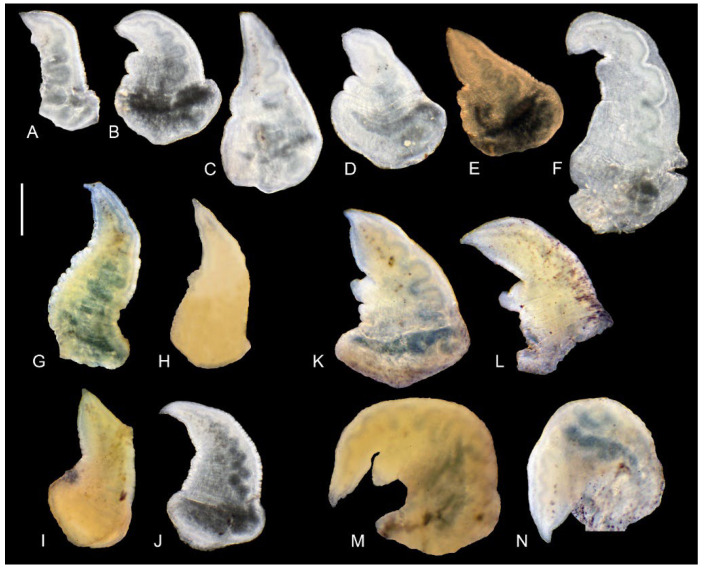
Penes of the studied taxa. (**A**–**F**)—*Dicle bilgini* n. sp. (locality 4); (**G**–**J**)—*Adaniya tufanbeyi* n. sp. (locality 1); (**K**–**N**)—*Adaniya ozbeki* n. sp. (locality 2). Scale bar: 200 μm.

**Figure 12 animals-15-02512-f012:**
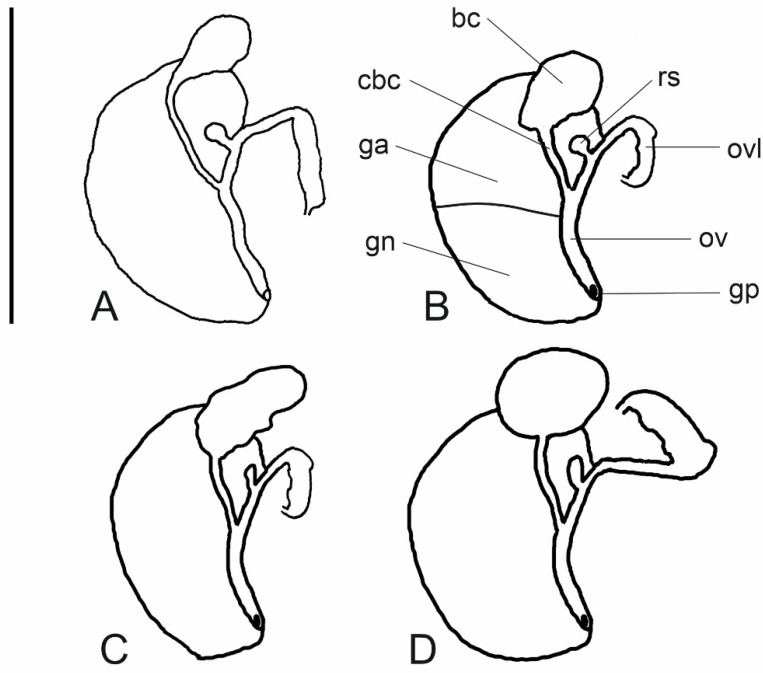
Female reproductive organs. (**A**)—*Adaniya tufanbeyi* n. sp. (locality 1); (**B**)—*Adaniya ozbeki* n. sp. (locality 2); (**C**)—*Adaniya cuneytsolaki* n. sp. (locality 3); (**D**)—*Anadoludamnicola ayhani* n. sp. (locality 3). Abbreviations: bc—bursa copulatrix; cbc—duct of bursa copulatrix; ga—albuminoid gland; gn—nidamental gland; gp—gonoporus; ov—oviduct; ovl—loop of (renal) oviduct; rs—receptaculum seminis. Scale bar: 1 mm.

**Figure 13 animals-15-02512-f013:**
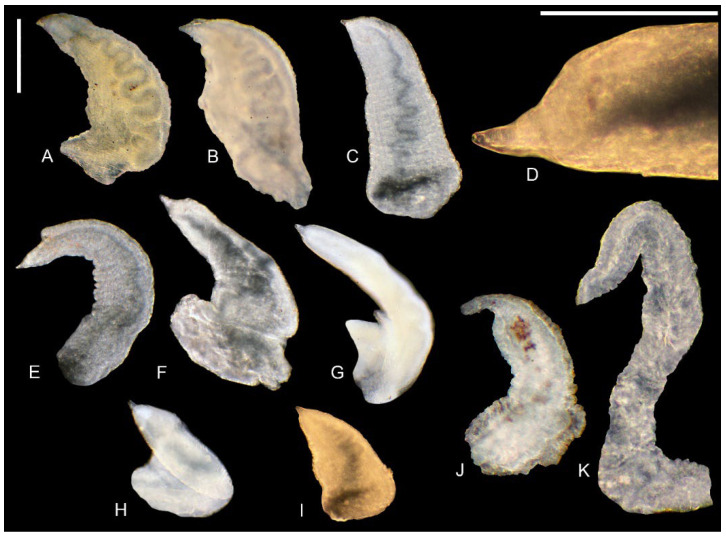
Penes of the studied taxa. (**A**,**B**)—*Adaniya cuneytsolaki* n. sp. (locality 3); (**C**–**I**)—*Sheitanok esmaae* n. sp. (locality 4); (**J**,**K**)—*Anadoludamnicola ayhani* n. sp. (locality 3). Scale bar: 200 μm and 100 μm (for (**D**) only).

**Table 1 animals-15-02512-t001:** Sampling locations of snails used in this study.

Id	Locality	Coordinates		Altitudes	Species (Extraction Number)
1	Sarız Çayı, Near Yamanlı Village, Tufanbeyli District, Adana Province.	38°10′21″ N	36°14′53″ E	1325 m	*Adaniya tufanbeyi* n. sp. (3K47)
2	Küçükçamurlu Spring, Göksun District, Kahramanmaraş Province.	37°52′32″ N	36°23′24″ E	1452 m	*Adaniya ozbeki* n. sp. (3K46)
3	Küp Şelalesi, Doğan Çayı, Aladağ District, Adana Province.	37°41′23″ N	35°30′12″ E	476 m	*Adaniya cuneytsolaki* n. sp. (3K42, 3K43); *Anadoludamnicola ayhani* n. sp. (3K44)
4	Spring in Finik Harabeleri (Finik Ruins), Eskiyapı, Güçlükonak, Şırnak. No.1.	37°24′26″ N	42°04′41″ E	443 m	*Dicle bilgini* n. sp. (3L24, 3L25); *Sheitanok esmaae* n. sp. (3L22)
5	Spring in Finik Harabeleri (Finik Ruins), Eskiyapı, Güçlükonak, Şırnak. No.3.	37°24′25″ N	42°04′21″ E	444 m	*Dicle bilgini* n. sp. (3L30, 3L31); *Sheitanok esmaae* n. sp. (3L28, 3L29)

**Table 2 animals-15-02512-t002:** COI sequences used for phylogenetic inference.

Species	GB Numbers	References
*Anadoludamnicola gloeri* Şahin, Koca & Yıldırım, 2012	OP096205	Delicado et al. 2024 [14]
*Arganiella pescei* Giusti & Pezzoli, 1980	MW553909	Delicado et al. 2021 [40]
*Belgrandia thermalis* (Linnaeus, 1767)	AF367648	Wilke et al. 2001 [41]
*Belgrandiella* cf. *kuesteri* (Boeters, 1970)	MG551325	Osikowski et al. 2018 [42]
*Chilopyrgula zilchi* Schütt, 1964	OP096298	Delicado et al. 2024 [14]
*Daphniola louisi* Falniowski & Szarowska, 2000	KM887915	Szarowska et al. 2014 [43]
*Ecrobia maritima* (Milaschewitsch, 1916)	KX355835	Osikowski et al. 2016 [44]
*Erosiconcha geldiayana* (Schütt & Bilgin, 1970)	MZ606137	Delicado & Gürlek 2021 [10]
*Falsipyrgula pfeiferi* (A. Weber, 1927)	EF379296	Wilke et al. 2007 [45]
*Fissuria boui* Boeters, 1981	AF367654	Wilke et al. 2001 [41]
*Graecoanatolica lacustristurca* Radoman, 1973	OP096234	Delicado et al. 2024 [14]
*Graecoanatolica pamphylica* (Schütt, 1964)	OP096236	Delicado et al. 2024 [14]
*Graecoarganiella parnassiana* Falniowski & Szarowska, 2011	MZ093454	Hofman et al. 2021 [46]
*Graecoanatolica tenuis* Radoman, 1973	OP096237	Delicado et al. 2024 [14]
*Grossuana angeltsekovi* Glöer & Georgiev, 2009	KU201090	Falniowski et al. 2016 [47]
*Hauffenia tellinii* (Pollonera, 1898)	KY087861	Rysiewska et al. 2017 [48]
*Iglicopsis butoti* Falniowski & Hofman, 2021	MW879273	Falniowski et al. 2021 [49]
*Islamia anatolica* Radoman, 1973	OP096253	Delicado et al. 2024 [14]
*Islamia bunarbasa* (Schütt, 1964)	OP096255	Delicado et al. 2024 [14]
*Islamia pseudorientalica* Radoman, 1973	OP096261	Delicado et al. 2024 [14]
*Islamia zermanica* Radoman, 1973	KU662362	Beran et al. 2016 [50]
*Kozanium torosum* Odabaşı & Falniowski, 2025	PV494927-PV494928	Odabaşı et al. 2025 [13]
*Pseudorientalia* sp., Samos Island	KJ920477	Szarowska et al. 2014 [51]
*Radomaniola curta* (Küster, 1853)	KC011814	Falniowski et al. 2012 [52]
*Sadleriana fluminensis* (Küster, 1853)	KF193067	Szarowska & Falniowski 2013 [53]
*Sheitanok* sp.	OP096317	Delicado et al. 2024 [14]
*Tefennia tefennica* Schütt & Yıldırım, 2003	OP096320	Delicado et al. 2024 [14]

**Table 3 animals-15-02512-t003:** Uncorrected p-distances between mOTUs shown in Figure 4. Numbers in italics and bold indicate distances within mOTU. mOTUs means molecular Operational Taxonomic Units.

	A	B	C	D	E	F	G	H	I
A	** *0.001* **								
B	0.097	** *-* **							
C	0.089	0.028	** *-* **						
D	0.096	0.040	0.025	** *0.002* **					
E	0.123	0.114	0.121	0.122	** *0.014* **				
F	0.133	0.125	0.129	0.132	0.071	** *-* **			
G	0.127	0.101	0.099	0.098	0.111	0.110	** *-* **		
H	0.112	0.108	0.108	0.111	0.109	0.116	0.058	** *-* **	
I	0.105	0.101	0.103	0.115	0.100	0.114	0.110	0.082	** *0.000* **

**Table 4 animals-15-02512-t004:** Shell measurements in mm. Measured variables: a—shell height; b—body whorl width; c—aperture height; d—spire height; e—aperture width; α—apex angle measured between the lines tangential to the spire; β—angle between the body whorl suture and the line perpendicular to the columella.

	a	b	c	d	e	α	β
***Dicle bilgini*** n. sp.							
holotype	1.83	1.11	0.89	0.56	0.73	73	15
3L24	1.78	1.08	0.86	0.51	0.71	77	13
3L25	1.67	1.08	0.82	0.42	0.71	77	11
3L30	1.73	1.06	0.84	0.48	0.78	76	13
1	1.82	1.15	0.85	0.54	0.78	73	14
2	1.87	1.22	0.95	0.41	0.86	79	11
3	1.74	1.18	0.88	0.38	0.81	83	11
4	1.77	1.12	0.85	0.45	0.74	75	13
M	1.78	1.13	0.87	0.47	0.77	76.63	12.63
SD	0.06	0.06	0.04	0.06	0.05	3.29	1.51
Min	1.67	1.06	0.82	0.38	0.71	73	11
Max	1.87	1.22	0.95	0.56	0.86	83	15
***Adaniya tufanbeyi*** n. sp.							
holotype	2.48	1.43	1.25	0.85	1.00	78	14
3K47	2.56	1.47	1.18	0.88	1.01	80	14
1	2.50	1.43	1.09	0.86	0.92	74	12
2	3.12	1.70	1.49	1.05	1.16	74	15
3	2.53	1.52	1.31	0.77	1.05	82	14
4	2.45	1.47	1.23	0.78	1.01	81	12
5	2.64	1.47	1.17	0.92	1.02	78	18
6	2.51	1.55	1.24	0.73	1.10	83	11
M	2.60	1.51	1.25	0.86	1.03	78.75	13.75
SD	0.22	0.09	0.12	0.10	0.07	3.41	2.19
Min	2.45	1.43	1.09	0.73	0.92	74	11
Max	3.12	1.70	1.49	1.05	1.16	83	18
***Adaniya ozbeki*** n. sp.							
holotype	2.32	1.53	1.28	0.56	1.11	84	13
3K46	2.28	1.51	1.22	0.55	1.10	88	15
1	2.16	1.43	1.15	0.53	1.03	91	9
2	2.42	1.63	1.21	0.51	1.16	83	9
3	2.44	1.53	1.29	0.62	1.13	85	11
4	2.28	1.53	1.33	0.47	1.11	89	9
5	2.27	1.56	1.25	0.52	1.16	88	9
M	2.31	1.53	1.25	0.54	1.11	86.86	10.71
SD	0.10	0.06	0.06	0.05	0.04	2.91	2.43
Min	2.16	1.43	1.15	0.47	1.03	83	9
Max	2.44	1.63	1.33	0.62	1.16	91	15
***Adaniya cuneytsolaki*** n.sp.							
holotype	1.53	0.87	0.65	0.45	0.62	83	13
3K42	1.83	1.03	0.83	0.36	0.72	80	10
3K43	1.80	0.98	0.84	0.52	0.66	80	15
1	1.67	0.94	0.72	0.44	0.62	85	10
2	1.47	0.92	0.74	0.34	0.62	89	12
3	1.57	0.92	0.72	0.39	0.66	86	14
4	1.54	0.93	0.72	0.37	0.64	86	15
5	1.73	0.98	0.81	0.38	0.71	83	11
M	1.64	0.95	0.75	0.41	0.66	84.00	12.50
SD	0.13	0.05	0.07	0.06	0.04	3.12	2.07
Min	1.47	0.87	0.65	0.34	0.62	80	10
Max	1.83	1.03	0.84	0.52	0.72	89	15
***Sheitanok esmaae*** n.sp.							
holotype	1.00	0.88	0.70	0.12	0.59	120	9
3L22	1.05	0.93	0.73	0.13	0.67	113	9
3L28	0.82	0.70	0.60	0.09	0.50	112	7
3L29	0.86	0.76	0.64	0.07	0.58	117	6
1	1.27	1.05	0.78	0.15	0.65	111	8
2	1.14	0.99	0.78	0.16	0.66	111	12
3	0.95	0.81	0.67	0.15	0.58	105	8
4	1.06	0.89	0.75	0.17	0.62	103	11
M	1.02	0.88	0.71	0.13	0.61	111.50	8.75
SD	0.15	0.12	0.07	0.04	0.06	5.61	1.98
Min	0.82	0.70	0.60	0.07	0.50	103	6
Max	1.27	1.05	0.78	0.17	0.67	120	12
***Anadoludamnicola ayhani*** n.sp.							
holotype	1.40	0.90	0.84	0.25	0.66	91	7
3K44	1.66	1.05	0.94	0.35	0.80	88	9
1	1.41	0.88	0.75	0.33	0.69	92	9
2	1.46	0.90	0.78	0.26	0.72	95	7
3	1.42	0.93	0.83	0.28	0.74	94	7
4	1.49	0.96	0.86	0.30	0.72	95	8
5	1.37	0.86	0.77	0.30	0.65	94	8
M	1.46	0.93	0.82	0.30	0.71	92.71	7.86
SD	0.10	0.06	0.07	0.04	0.05	2.56	0.90
Min	1.37	0.86	0.75	0.25	0.65	88	7
Max	1.66	1.05	0.94	0.35	0.80	95	9
** *Kozanium torosum* **							
holotype	1.43	0.97	0.82	0.25	0.72	90	9
3i83	1.67	1.04	0.83	0.34	0.74	86	11
anatomy	1.45	0.94	0.75	0.29	0.70	89	11
3J83	1.32	0.93	0.82	0.19	0.72	94	8
3J82	1.39	0.95	0.81	0.26	0.69	91	8
M	1.45	0.97	0.81	0.27	0.71	90.00	9.40
SD	0.13	0.04	0.03	0.06	0.02	2.92	1.52
Min	1.32	0.93	0.75	0.19	0.69	86	8
Max	1.67	1.04	0.83	0.34	0.74	94	11

**Table 5 animals-15-02512-t005:** The effects of species on biometrical variability in individuals.

Simple Term Effects:					Conditional Term Effects:				
Name	Explains%	Pseudo-F	P	P (adj)	Name	Explains%	Pseudo-F	P	P (adj)
Adatuf	31.7	22.7	0.001	0.003	Adatuf	31.7	22.7	0.001	0.0012
Sheesm	30.2	21.2	0.001	0.003	Sheesm	21	21.4	0.001	0.0012
Adaozb	12.6	7	0.004	0.008	Adaozb	14.5	20.7	0.001	0.0012
Anaayh	10.5	5.7	0.01	0.015	Anaayh	9.5	18.9	0.001	0.0012
Dicbil	4.9	2.5	0.111	0.111	Koztor	4.9	12.2	0.001	0.0012
Adacun	4.7	2.4	0.108	0.111	Dicbil	1.3	3.5	0.047	0.047
Koztor	3.1	1.6	0.204	0.204					

**Table 6 animals-15-02512-t006:** Comparative table summarizing diagnostic features of the novel described species.

Species	Shell, Shape, and Size	Radula (Central Tooth)	Female Reproductive Organs	Penis Morphology
*Dicle bilgini* n. sp.	Ovate-conic, broad; up to 1.87 mm; spire moderately high; ~4 whorls; aperture broadly ovoid; umbilicus moderately broad (Figure 9A–E; Table 4)	Formula (5)4–1–4(5); median cusp ~ 3× longer than adjacent; basal tongue narrowly V-shaped; basal cusps present (Figure 10H)	Cylindrical, bulky bursa copulatrix; one distal receptaculum seminis;	Simple, gradually tapering or broadly triangular; small left-side outgrowth; blunt tip with small papilla (Figure 11A–F)
*Adaniya tufanbeyi* n. sp.	Ovate-conic, broad; up to 3.12 mm; spire moderately high; umbilicus slit-like (Figure 9F,G; Table 4)	Formula (10)9–1–9(10); two pairs basal cusps; massive cutting edge (Figure 10F)	Small bursa copulatrix, duct not sharply demarcated (Figure 12A); large distal receptaculum seminis;	Broadly triangular; simple; very small left-side outgrowth (Figure 11G–J)
*Adaniya ozbeki* n. sp.	Trochiform, broad; up to 2.44 mm; spire low; umbilicus broad (Figure 9H,I; Table 4)	Formula (6)5–1–5(6); one pair basal cusps; massive cutting edge (Figure 10C,D)	Big oval bursa copulatrix with short, broad duct; spherical receptaculum seminis (Figure 12B)	Broadly or wide triangular nonglandular outgrowth on the medial left side (Figure 11K–N)
*Adaniya cuneytsolaki* n. sp.	Small, ovate-conic, narrow; up to 1.83 mm; spire low–moderately high; umbilicus broad trough (Figure 9J–L; Table 4)	Formula (5)4–1–4(5) or 4–1–4; few, large sharp cusps; basal cusps present (Figure 10G)	Elongated sac-shaped bursa copulatrix with short, broad duct; sac-shaped receptaculum seminis (Figure 12C)	Broadly triangular; no outgrowth (Figure 13A,B)
*Sheitanok esmaae* n. sp.	Valvatiform (depressed trochiform), broad; up to 1.27 mm; spire extremely low; umbilicus moderately broad, open (Figure 9M–R; Table 4)	Formula 5–1–5; basal cusps 1–1; cusps blunt (Figure 10E)	Narrow tubular bursa copulatrix; one small distal receptaculum seminis	Gradually tapering; simple; large terminal papilla with thin chitin layer (Figure 13C–I)
*Anadoludamnicola ayhani* n. sp.	Ovate-conic, broad; up to 1.66 mm; spire rather low; umbilicus broad (Figure 9S,T; Table 4)	Formula (6)5–1–5(6); no basal cusps (Figure 10A,B)	Spherical bursa copulatrix with a distinct short duct; sac-shaped receptaculum seminis (Figure 12D)	Strap-like, narrow; pointed tip (Figure 13J,K)

## Data Availability

All the sequences are deposited in GenBank accession numbers PX127888-PX127899.

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
