# Peer review of "Entirely Anatolian Hydrobiid (Caenogastropoda–Truncatelloidea–Hydrobiidae) Clade Revisited: Two More New Genera and Six New Speciesâ€"

_animals, 2025, doi:10.3390/ani15172512_

Round 1
Reviewer 1 Report
Comments and Suggestions for Authors
Review of “Entirely Anatolian hydrobiid (Caenogastropoda: Truncatelloidea: Hydrobiidae) Clade Revisited: Two More New Genera and Six New Species” by JaszczyÅ„ska et al. 2025
General comments on the manuscript
The present work is a continuation of previous studies on the diversity of the Hydrobiidae endemic to Anatolia, Türkiye. The aim of the study was to clarify the relationships of the new records with the previously collected species and to describe the new taxa.
To achieve these goals, the authors used new generated Cytochrome oxidase subunit 45 I (COI) sequences, as well as examined morphological shell features and the soft anatomy of the snails. This combination was necessary because the many species of the region can hardly be separated morphologically. The presented new species (and genera) are, at least according to morphology and anatomy, clearly distinguishable from those previously described, all results are also well supported statistically.
The paper is properly organized and clearly written, although I'm not a native speaker, the English seems to me to be well. Also, the title and abstract are informative and adequate for the content of the manuscript, although the name of the new species may appear in the manuscript somewhat earlier (in the abstract?). I did not find any errors in the descriptions, diagnoses, and taxonomy (except for the fact that the holotypes do not have a collection number!). The very detailed methodology is reproducible and appropriate for the objectives of the manuscript.
Generally, the authors bring sufficient arguments to prove that the collected populations are new independent lineages. The relationships to sister species from the region are also discussed in detail, although not in the discussion section, but in the “Taxonomic Remarks”. The conclusions of the authors are strongly supported by the data and analysis.
The literature is cited adequately, and most current references are included. The paper contain a high proportion of the cited references belong to authors. It could be because one of the authors is a well-known expert in the field with numerous publications... Nevertheless, I could not find any superfluous citations.
I only found few minor issues in the manuscript (see PDF file).
Reviewer 1

Author Response
I have done all the changes
Reviewer 2 Report
Comments and Suggestions for Authors
The manuscript is devoted to the taxonomy of the family Hydrobiidae, one of the complicated families of Truncateloida with a huge number of taxa, often difficult to distinguish by shell characteristics. The authors described 6 new species and 2 new genera from five Anatolian streams using morpho-anatomical, biometric, and molecular phylogenetic methods in an integrative manner. It is interesting that the three described species of the genus Adaniya differ in the teeth of the radula, which is quite rare in species of the same genus living in different localities. I was also interested in the fact that two allopatric species have the same shell shape, but they are not closely genetically related.
I liked the article, and I have no major comments. But I have some minor requests
Table 3 p-distances…. Maybe better to write "uncorrected p-distances"?
All described species have a smooth operculum on both surfaces. But what kind of operculum does every species have? Round, oval, or elongate-ellipsoidal? Paucispiral with submarginal nucleus or other?
If possible, please clarify
Line 317
>Head, tentacles and mantle intensively pigmented, no ctenidium.
Does the species have an unpigmented ctenidium? More precisely, please.
I hope that the authors will soon compile a key to identify the species of the Hydrobiidae family they studied.
Author Response
All changes done
Reviewer 3 Report
Comments and Suggestions for Authors
Review for the paper “Entirely Anatolian hydrobiid (Caenogastropoda: Truncatelloidea: Hydrobiidae) clade revisited: two more new genera and six new species” by Aleksandra JaszczyÅ„ska, Sebastian Hofman, Deniz Anıl OdabaÅŸi, İhsan Ekin, Serdar Polat, Ioan Sîrbu, Andrzej Fryderyk Falniowski submitted to “Animals”.
The authors of this research paper conducted an analysis of the Hydrobiidae family in Anatolia, focusing on endemic species and their relationships to European snails. In their investigation, the authors collected samples from five springs across Anatolia. They described the shell morphology, biometry, radulae, and soft part pigmentation of these snails, along with illustrations of their female reproductive organs and penes. They used cytochrome oxidase subunit I sequences to explore phylogenetic relationships and assess the distinctness of the species within this group. The results of this study revealed the existence of six species that were new to science, signifying a significant contribution to understanding Hydrobiidae diversity in the region. The results of this study may have important implications for future research in malacology and the conservation of freshwater biodiversity in Anatolia.
Some revisions are needed to improve the clarity of the paper.
Recommendations.
Introduction.
L 56-58. The authors should provide an overview of the research that has been done on hydrobiid fauna in other Mediterranean countries for comparison.
L 65. The authors should explain what is meant by the "west-south-east pattern of divergence".
L 66. The authors should discuss the ecological or historical factors that explain why these genera have a circum-Mediterranean distribution.
The rationale for this study should be explained more clearly. The authors should report why these findings are significant for conservation, evolutionary biology, or biodiversity assessment in Turkiye and the broader Mediterranean region.
Materials and Methods.
L 82-83. The authors should explain the methodology of "sieve collection" in more detail. What type of sieve was used? How were the snails distinguished from debris or sediment? Were specific protocols followed to ensure uniform sampling across locations?
L 95. The authors should provide elevation data for locality 2.
L 101-102. The authors should explain what is meant by "springs on the right-hand side" Is this a geographical or hydrological reference? How was the spring water differentiated from the stream water?
L 116-117. Although the authors give a reference, it would be useful to specify which seven biometric parameters were measured. The authors should place figure 4 in the "Materials and Methods".
L 155. The authors should explain which genetic distance model was used for the calculation.
Results.
Table 3. The authors should explain the meaning of the bold font.
L 246. Change "Figre 8" to "Figure 8".
It would be useful to include a comparative table summarizing diagnostic features of these species.
Discussion.
L 650-653. The authors should discuss why two species delimitation techniques failed to identify the same species structure as the morphological evidence.
All references are appropriate because there are no alternatives for the authors' self-citations.
Author Response
all changes done